# Probiotics in Allergic Rhinitis Management: Is There a Positioning for Them?

**Giorgio Ciprandi** [1,*] and **Maria Angela Tosca** [2]

1   Allergy Clinic, Casa di Cura Villa Montallegro, 16100 Genoa, Italy
2   Allergy Center, Istituto Giannina Gaslini, 16100 Genoa, Italy
*   Correspondence: gio.cip@libero.it

**Abstract:** Allergic rhinitis (AR) is a widespread medical condition affecting up to 40% of the general population. Type 2 inflammation determines typical nasal symptoms. In addition, gut and respiratory dysbiosis are present in AR patients. Probiotics have several beneficial effects on immunity, inflammatory pathways, and anti-infective properties. Namely, probiotic supplementation could restore immune response, promote eubiosis, and switch off inflammation. Thus, probiotics have also been investigated in AR. In addition, there is accumulating evidence that some specific strains of probiotics may improve AR. Five meta-analyses on probiotics in AR management were consistently published in the first half of 2022. The conclusions, although not definitive, argue for the possible use of probiotics as part of an add-on strategy in managing patients with allergic rhinitis.

**Keywords:** oral probiotics; allergic rhinitis; dysbiosis; inflammation; immunity; add-on therapy

## 1. Introduction

Allergic rhinitis (AR) is a common medical condition affecting up to 40% of the general population [1]. A type 2 immunity determines eosinophilic inflammation that, in turn, elicits typical nasal symptoms [2]. Type 2 immunity is eminently characterized by polarization of innate and adaptative B and T cells, increased production of type 2 cytokines, including interleukin-4 (IL-4), IL-5, and IL-13, and impaired function of allergen-specific T regulatory cells (Tregs) [3,4]. This immunologic derangement promotes allergic inflammation, characterized by an abundant eosinophilic infiltrate and the presence of mast cells [5]. The mast cells are activated by allergen exposure and release pro-inflammatory mediators, including histamine [6]. These mediators interact with specific receptors and, consequently, are responsible for the appearance of typical AR symptoms: nasal itching, sneezing, rhinorrhea, and nasal congestion [7].

On the other hand, allergic rhinitis can present with different phenotypes and endotypes and is often associated with numerous comorbidities, especially with the involvement of the bronchi and conjunctiva [8]. In addition, a type 1 immunity defect follows from the expansion of the type 2 polarization; thus, allergic subjects are prone to contracting many infections [9].

In light of all this evidence, much attention has been paid in recent years to the pathogenetic role in AR exerted by the microbiota [10]. There is documented data that patients with AR have dysbiosis at the nasal level and in the gut [11,12]. Therefore, there is convincing evidence that deranged microbiota promotes, amplifies, and maintains the pathophysiologic mechanisms involved in AR [13]. Early antibiotic abuse is associated with an increased prevalence of allergic disorders [14].

As a result, new strategies in managing AR patients have been advanced to restore the microbial population's imbalanced composition in a qualitative and quantitative sense [15,16]. In addition, probiotics have been assigned a prominent role as potential contributors to eubiosis [17].

## 2. The Rationale for Probiotic Use in Allergic Rhinitis Management

There is a well-established awareness that the prevalence of allergic diseases has been impressively raised in the last decades. Consequently, several attempts have been made to explain this important phenomenon. In this context, the hygiene hypothesis has garnered much acclaim and has been mainly studied [18]. The observation has sustained the so-called allergy epidemics in which poor hygienic conditions protect from allergies while "clean" settings arrange for allergies. Namely, adequate microbiota biodiversity guarantees a correct maturation of the immune system in infants [19]. Contrarily, the impaired composition of microbiota facilitates poor immunity maturation and favors allergy [20]. Moreover, newborns display a type 2 immunity as this immune disposition protects the fetus from the potential maternal rejection as recognized non-self.

Allergic subjects present dysbiosis and impaired biodiversity in the gut and target organs [21]. Dysbiosis harms AR as it promotes allergic inflammation. However, there is evidence that some environmental factors may contribute to eubiosis [22]. In particular, exposure to microbial fragments (e.g., farming settings) promotes balanced microbiota preventing allergy [23].

Moreover, a gut-airways axis permits the crosstalk between the digestive tract and respiratory mucosa. In other words, enteric microbiota significantly affects respiratory microbiota and vice versa [24]. Subjects with AR present a dysbiosis associated with inflammatory pathways and symptom severity [25]. Therefore, the combination of all this evidence suggests the possibility of manipulating the immune system to return a physiological response in AR. In this regard, probiotics have been proposed as an ideal candidate to counteract dysbiosis in allergic rhinitis [26]. Probiotics are "live microorganisms which confer a beneficial effect on the host", as proposed by the World Health Organization [27].

Probiotic supplementation may exert manifold benefits due to its complex mechanism of action [28]. Probiotics represent the current strategy to heal dysbiosis revitalize biodiversity, and balance microbiota. However, the mechanisms of action are complicated and still not completely defined. The detailed description of the probiotics mechanisms of action goes beyond the purpose of the present review. Consequently, a summary of the concept will be presented. Many recent reviews are available [29–41].

Briefly, probiotics act on innate immunity through the toll-like receptor, switching off the antigen-presenting machinery. The activation of innate immunity displays several positive effects, as schematically represented in Figure 1. However, some effects on immune results have to be presumed as the present evidence derives from small studies.

First, probiotics may down-regulate type 2 immunity by reducing allergen-specific IgE and "pro-allergic" cytokine production; contextually, probiotics stimulate allergen-specific IgG and IgA production. Consistently, probiotics may expand type 1 immunity, increasing the production of "protective" cytokines (IFN-$\gamma$, IL-6, and TNF-$\alpha$). As a result, probiotics make allergic subjects less prone to infections and also promote the synthesis of anti-infective agents, such as bacteriocin and secretory IgA. In addition, probiotics may dampen the inflammatory cascade by stimulating the production of anti-inflammatory mediators, including short-chain fatty acids (mainly $\omega$3), that reduce cellular infiltrate. Finally, probiotics may restore the function of allergen-specific Tregs and consequently increase the production of regulatory cytokines (IL-10 and TGF-$\beta$).

Therefore, there is evidence that probiotics could confer some beneficial effects to allergic subjects rebalancing their immunity and reinforcing mucosal defense against pathogens. However, more methodologically robust studies are needed to confirm these possible effects of probiotics.

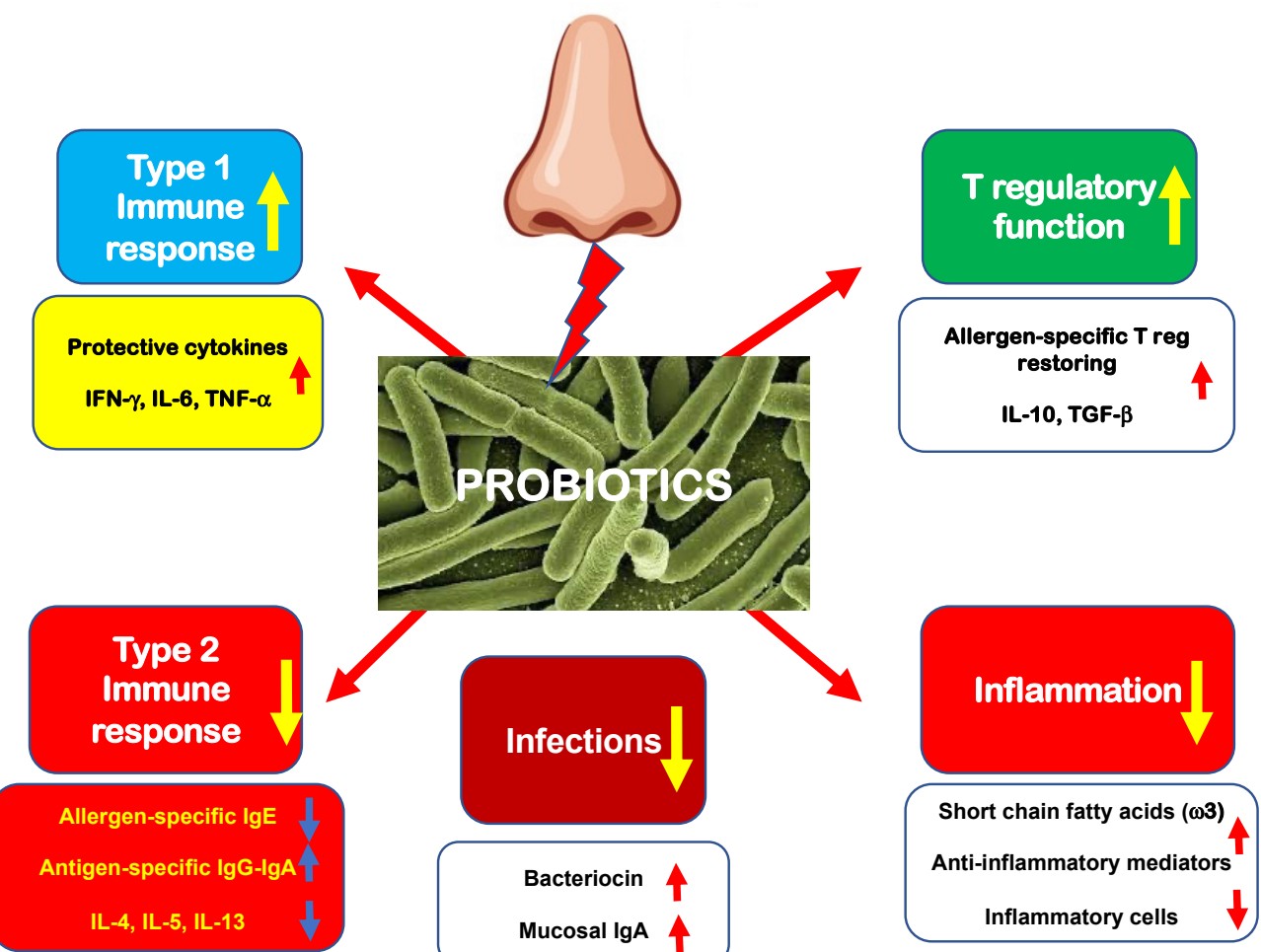

**Figure 1.** Schematic representation of the possible probiotic mechanisms of action in allergic rhinitis.

### 3. Recent Meta-Analyses on Probiotics in Allergic Rhinitis

The interest in probiotics in managing AR patients is intriguing, as the possibility of manipulating the immune system through "good" bacteria is fascinating. The publication of as many as five meta-analyses in the first half of 2022 alone supports this scientific interest. The enormous potential of this subject can only justify such a broad and contextual claim. Therefore, we will now discuss these recent meta-analyses published this year, excluding past meta-analyses on this topic.

Chen and coworkers performed a meta-analysis of probiotics for respiratory allergies in children [42]. These researchers included 15 randomized controlled studies, including 1388 patients. The analysis showed that probiotics significantly improved quality of life and symptom severity. However, the authors concluded that further research must thoroughly explain probiotics' fundamental mechanisms of action. These conclusions also suggested that it is entirely incorrect to gather very different probiotics, but the single probiotic strain should be separately analyzed. Only reliable outcomes can be obtained when the analysis focuses on a single strain. Therefore, analyzing the data concerning a single probiotic strain is necessary. Furthermore, each strain may exert specific activities depending on genetic, adaptative, immunological, and metabolic characteristics that make every strain unique,

with potential specific efficacy and safety. Therefore, probiotic strains with proven benefits for a particular disease have to be selected.

Luo and colleagues included 28 studies on patients with AR [43]. The meta-analysis showed that probiotics significantly reduced symptom severity and improved quality of life. The analysis also demonstrated that probiotics reverted the ratio of T helper cell 1/T helper cell 2 toward a physiologic type 1 polarization. However, probiotics did not change the allergen-specific IgE levels. Based on these results, therefore, these authors concluded that probiotic supplementation ameliorated nasal symptoms and improved the quality of life of AR patients. However, the analysis detected a high heterogeneity in some outcomes. Consequently, doctors should cautiously use probiotics to treat AR.

Yan and coworkers performed another meta-analysis of probiotics in the AR treatment [44]. These investigators selected only randomized controlled trials (RCT). Globally, the analysis included 30 RCTs, enrolling 2708 patients. The analysis showed that the global and nasal scores of rhinitis quality of life significantly improved in probiotics-treated patients and the nasal symptom severity. However, probiotics did not affect eosinophil count, ocular symptoms, and total and allergen-specific IgE levels. Therefore, the authors concluded that probiotics improved AR patients' quality of life and symptoms. However, also this meta-analysis outlined limited evidence of the study results, heterogeneity of methods, and difference in findings. So, the authors concluded that there is a need to perform other high-quality studies on this issue.

Farahmandi and colleagues performed another meta-analysis considering only RCT of probiotic treatments in AR patients [45]. This analysis identified 13 RCTs. The analysis demonstrated that eight of nine probiotic strains diminished at least one AR symptom. In particular, *L. paracasei* Lp33 and *L. rhamnosus* GG did not significantly reduce nasal symptom severity. Therefore, the analysis highlighted a slight effect of some probiotic strains on some clinical and immunological parameters. Given that there was a diversity of outcome measurements and the number of RCTs was insufficient, the researchers concluded that further studies are required. A similar methodology is requested to compare the effects better.

Iftikhar and co-authors performed a systematic review of systematic reviews about the role of probiotics in AR patients [46]. These researchers screened 419 articles, but only three systematic reviews were eligible for the review. The review reported that probiotics in managing AR patients improved quality of life, total nasal symptom and ocular symptom scores, and total daily symptom score. In addition, probiotics affected the Th1/Th2 ratio. However, probiotics did not change some immunological parameters, including total and allergen-specific IgE, IL-10, IFN-$\gamma$, and eosinophil. The authors concluded that there is considerable evidence supporting the probiotics' usefulness in the AR treatment, but further RCTs are desirable to address the unmet needs.

In addition, this year Sun and coworkers published a meta-analysis regarding the preventive effects of probiotics against allergic disorders, mainly concerning eczema [47]. The authors identified five available studies about AR. The analysis reported no significant difference in preventing AR. However, it has to be noted that this analysis considered only the primary prevention of AR. This context is very different from the scenario in patients with established AR.

However, it should be noted that the four meta-analyses on the use of probiotics in the treatment of AR published this year included different numbers of studies also concerning the age of patients.

In addition, these meta-analyses analyzed different outcomes, and the selection and evaluation criteria were not all the same. These discrepancies can be easily observed when comparing meta-analyses published on the same topic. The results obtained from a meta-analysis are a function of the studies included, and the parameters analyzed. Each meta-analysis necessarily has peculiarities that other meta-analyses on the same topic cannot share. All the more so, these considerations apply to a meta-analysis of meta-analyses, which by definition represents a more complex and articulated work. What should be

considered attractive, however, is that all of these meta-analyses, while presenting even important differences, reached results that in all cases support some benefit exerted by probiotics in the management of RA. We can thus glimpse how inherent differences may translate into some uniformity of results.

However, we must also consider that there are currently many limitations for many studies, so these results must still be regarded as preliminary and not definitive regarding the use of probiotics in allergic rhinitis.

We still have to consider that most studies have clinical parameters, symptoms, and quality of life outcomes. In contrast, few studies have investigated immunological aspects. In addition, mechanistic studies are relevant to documenting probiotics' actual mechanism of action. All these considerations should therefore be taken into account while addressing this issue.

Therefore, these meta-analyses allow us to conclude that there is currently not yet sufficiently conclusive evidence to support the use of probiotics alone in treating RA. However, some benefits are evident, especially for some specific strains, although more studies conducted with rigorous and robust methodology are needed.

However, a fundamental concept must be stressed: probiotics are not drugs and cannot be considered symptomatic treatments for allergic rhinitis. Instead, they should be regarded as part of an adjunctive strategy to make the most of their potential to manipulate the immune response.

Indeed, there is a need to obtain new studies that investigate in detail the mechanism of action of probiotics on the immunological, inflammatory, and microbiological components. In addition, these future studies should consider all those factors that may influence the in vivo effects of antibiotics. In particular, it is presumable that many drugs may interfere with the colonization and metabolism of probiotics. The list would probably be long, but antibiotics, PPIs, laxatives, and other natural products that may modify the gut microbiota and biodiversity should be included. Of course, diet also plays a crucial role in the composition of the microbiota and in affecting the efficacy of a probiotic; think of the effects of fiber, which acts as a prebiotic, dietary lipids and proteins, and alcoholic beverages. Again, physical and/or psychological stress can undoubtedly influence the effects of probiotics.

Another aspect that deserves proper consideration in the interpretation of the results is offered by genetics. It is well known that there are numerous polymorphisms regarding receptors for bacterial fragments, particularly Toll-like receptors, but also other ligands, including TRPA1. We could even hypothesize a new branch of genetics to go alongside pharmacogenetics and pharmacogenomics, that is, to think of possible "probioticogenetics" and "probioticogenomics". Finally, all these considerations converge toward a more general concept of responders and non-responders: a well-known concept in pharmacology.

Finally, aspects related to the probiotic product must also be considered, especially its composition, number of live bacteria, their viability over time, and stability of the strain.

Therefore, we can say that numerous factors can influence the effects of probiotics in vivo.

Last but not least, the safety of probiotics also needs proper attention, especially in some particular categories of subjects, for example, subjects with immunodeficiencies.

In conclusion, in the future, we expect to acquire new studies that are methodologically appropriate and inclusive of this information.

## 4. New Evidence in the Literature

Four more studies on the use of probiotics in the treatment of AR have been published in the past year, which were not covered by the meta-analyses mentioned above. Therefore, they will be presented here to provide updated knowledge about this issue.

The first study aimed to investigate the probiotic prophylactic treatment in children with AR [48].

The tested compound was a probiotic mixture including *Bifidobacterium animalis* subsp. *Lactis* BB12 and *Enterococcus faecium* L3. The outcome measures were nasal symptom scores and the need for rescue medication. The study included 250 children and adolescents. Subjects were stratified into two groups: actively treated (150) or control (100). The probiotic mixture was administered three months before the pollen season. Patients treated with probiotics experienced less severe symptoms than controls. Consistently, probiotic-treated children used less symptomatic drugs than controls. Consequently, the investigators concluded that the mixture containing BB12 and L3 exerted significant preventive effects on the AR course.

The second study explored the effects of a commercial probiotic preparation containing four strains: *Lactobacillus acidophilus* LA02, *Lactobacillus delbrueckii* LDB01, *Lactobacillus rhamnosus* LR04, and *Streptococcus thermophilus* FP04 [49].

Twenty-eight AR patients took the probiotic mixture for 60 days; they were evaluated at baseline, after the treatment, and during a two-month follow-up. The probiotic product significantly reduced rhinitis total symptom scores and visual analog scale. Consistently, probiotics reduced the peripheral eosinophils, the levels of type 2 cytokines IL-4 and IL-5, and increased biodiversity in stool microbiota composition. Interestingly, the microbiota changes correlated with clinical and immunological parameters. Consequently, the study concluded that this probiotic preparation could be envisaged as a helpful add-on strategy in AR patients. The outcomes provided by this study were particularly impressive, mainly concerning the cytological and immunological results. It has to be noted that such positive results were not observed by most other studies exploring these aspects. However, it should be emphasized that this study should be considered a preliminary study conducted on a small sample of patients. Therefore, these results, which are undoubtedly attractive, need confirmation by studies conducted with a more robust and appropriate methodology.

Another study evaluated an intranasally administered probiotic assemblage containing *Lactobacillus rhamnosus* SP1, *Lactobacillus paracasei* 101/37, and *Lactococcus lactis* L1A, compared with placebo [50].

Participants were subdivided into two groups: actively treated (12) and placebo-treated (12). Treatment lasted three weeks. Outcome measures were quality of life scores, total nasal symptom score, peak nasal inspiration flow, fractional exhaled nitric oxide, and cytokines assay. Cultures were also performed to detect the colonization of strains. Unfortunately, the only significant result concerned a slight decrease in IL-17. Therefore, the authors concluded that the topical administration of a probiotic mixture was ineffective in AR patients. Probably, the duration and the topical route were inadequate to observe benefits.

Finally, an Australian study investigated the effects of a probiotic drink ("NC-Seasonal-Biotic") containing four strains: *Lactobacillus reuteri* GL 104, *Lactobacillus plantarum* LPL28, *Lactobacillus rhamnosus* MP108, and *Bifidobacterium lactis* BI04, and a fructooligosaccharide as prebiotic, compared with placebo [51]. Forty patients concluded the 10–12-week intervention period. The active group showed a significant improvement in clinical parameters and consistent quality of life scores. In addition, the probiotic drink restored the Th1/Th2 ratio.

These recent studies, therefore, added new proof evidencing that probiotics could help manage patients suffering from AR. However, two main lessons have to be acknowledged. First, the probiotic supplementation duration should be adequate to allow colonization and completion of the immunological mechanism of action. Second, the effectiveness depends on the specificity of single strains.

Moreover, the outcomes reported in AR were substantially consistent with what has been reported in the asthma model [52–54].

## 5. Conclusions

Evidence in the actual literature suggests that probiotics could be a potential add-on strategy in managing patients with AR. However, no definitive recommendations can still be defined now.

**Author Contributions:** Conceptualization, G.C. and M.A.T.; methodology, G.C.; writing—original draft preparation, G.C.; writing—review and editing, M.A.T. All authors have read and agreed to the published version of the manuscript.

**Funding:** This research received no external funding.

**Institutional Review Board Statement:** Not applicable.

**Informed Consent Statement:** Not applicable.

**Data Availability Statement:** Not applicable.

**Conflicts of Interest:** The authors declare no conflict of interest.

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
