# Peer review of "Probiotics in Allergic Rhinitis Management: Is There a Positioning for Them?"

_allergies, doi:10.3390/allergies2030011_

Round 1
Reviewer 1 Report
This paper does not provide sufficient evidence-based literature to support the authors' conclusion. All the meta-analyses do not support the effect of probiotics on AR symptoms. Additionally, although 3 out of 4 of the recent literature seem to show that probiotics have a positive effect on reducing patients' AR symptom scores, the sample size are too low to be representative, given the high prevalence of people with AR worldwide.
Author Response
This paper does not provide sufficient evidence-based literature to support the authors' conclusion. All the meta-analyses do not support the effect of probiotics on AR symptoms. Additionally, although 3 out of 4 of the recent literature seem to show that probiotics have a positive effect on reducing patients' AR symptom scores, the sample size are too low to be representative, given the high prevalence of people with AR worldwide.
R We would thank the Reviewer for the comments. Unfortunately, we cannot agree with the Reviewer on these statements since all 4 meta-analyses and the meta-analysis of meta-analyses reported an effect of probiotics on nasal symptoms. We report below what is stated in the texts.
Regarding sample size, we are unclear about the comment regarding whether the sample size of a study should take into account the prevalence of a disease. The sample size should be calculated according to the expected outcomes in the specific study and not obviously the prevalence of disease.
Chen = …Significantly greater reduction in NSS (SMD =−1.43, 95% CI [ −1.63,−1.23], P < 0.01) and ESS (total MD=−1.67, 95% CI [ −1.79,−1.55], P < 0.01) were observed in probiotics group compared to control group…
Farahmandi = …Overall, 8 of 9 probiotic types alleviated at least one clinical symptom of allergic rhinitis…
Luo = …The results showed that probiotics significantly relieved allergic rhinitis symptoms (standardized mean difference [SMD], −0.29, 95% confidence interval (CI) [−0.44, −0.13]; p = 0.0003, I2 = 89%)…
Yan = …Meta-analysis results showed that the RTSS nasal scores (MD = −1.96; P = 0.02) significantly improved in the probiotic group when compared with those in the placebo group….
Iftikhar = …Probiotics in the treatment of AR have been shown to improve quality of life, the total nasal and ocular symptom scores, the daily total symptom scores and Th1/Th2 ratio…
On the other hand, we agree with the Reviewer about the concept that the present knowledge concerning the role of probiotics in AR cannot be considered definitive. Indeed, there is a need to perform new robust studies, including mechanistic ones, to obtain conclusive and convincing outcomes.
Reviewer 2 Report
The review is a thorough and up to date review of the state of use of probiotics in the treatment for allergic rhinitis and I believe the authors comments -- whilst initially teasing the reader that there may be enough evidence to use; come to a logical conclusion that there is a lot of heterogeneity in the study.
It is striking that of the 4 new systematic reviews looking at probiotics for the treatment of AR have quite different numbers of studies included. Chen was paediatric, but the Lou and Yan papers had 28 and 30 studies respectively. It would be interesting for the reader to know if they included similar studies. The systematic review of systematic reviews by Iftikhar talks about a change in Th1/2 balance. This is important and should be discussed more in detail (statistical relevance) as the previous 3 reviews only reported changes in symptoms/QoL without an immunological mechanism of action.
A systematic review of probiotics in prevention or AR and did not find any change.
The paper then moves to 4 new studies on the treatment of AR. Only one is of a suitable size. One paper of 28 patients..
- Torre, E.; Sola, D.; Caramaschi, A.; Mignone, F.; Bona, E.; Fallarini, S. A Pilot Study on Clinical Scores, 320 Immune Cell Modulation, and Microbiota Composition in Allergic Patients with Rhinitis and 321 Asthma Treated with a Probiotic Preparation. Int. Arch. Allergy Immunol. 2022, 183, 186-200
Makes very strong immunological findings which had not been found in many other studies that the authors have highlighted. Specifically that probiotics reduced the peripheral eosinophils, the levels of type 2 cytokines IL-4 and IL- 5 in addition to the changes in clinical scores. This needs some commentary about the fact that this is a small study. No comment was made about the timing of this study. It would be important to know if immunological differences correlated with clinical symptoms.
In the final statements
“The actual literature evidence reasonably seems to suggest that probiotics could be a profitable add-on strategy in managing patients with AR. However, no definitive recommendations can still be defined now.”
I think a better word than “profitable” could be used.
The mechanisms picture is nice, but there is only data from small studies to support the immunological changes that is still not convincing.
As a comment
There are many co-factors that might influence probiotics in any disease such as antibiotic use, PPIs, diet (fibre, lipids, fast food, protein, fructose) and even stress. Add on top of this, SNPs in Toll4 receptors and other bacterial ligands like TRPA1, there is always going to massive heterogeneity. In probiotic studies, it may be more beneficial to look at responders and non-responders to better elucidate how and if these agents are working. Further to this we have strain variation as well as number of bacteria and viability of the probiotics used
The authors need not include those thoughts and indeed, as an expert in the area - I am sure Professors Ciprandi and Tosca could have many other factors to add.
Overall – as mentioned it is an update from a highly respected author. I would like to see commentary about possible papers that the recent reviewers missed. The review answers thew question posed in the title.
Author Response
The review is a thorough and up to date review of the state of use of probiotics in the treatment for allergic rhinitis and I believe the authors comments -- whilst initially teasing the reader that there may be enough evidence to use; come to a logical conclusion that there is a lot of heterogeneity in the study.
R Many thanks for this comment.
It is striking that of the 4 new systematic reviews looking at probiotics for the treatment of AR have quite different numbers of studies included. Chen was paediatric, but the Lou and Yan papers had 28 and 30 studies respectively. It would be interesting for the reader to know if they included similar studies. The systematic review of systematic reviews by Iftikhar talks about a change in Th1/2 balance. This is important and should be discussed more in detail (statistical relevance) as the previous 3 reviews only reported changes in symptoms/QoL without an immunological mechanism of action. A systematic review of probiotics in prevention or AR and did not find any change.
R Many thanks for this comment. We expanded the discussion concerning these aspects. In addition, we changed the title and text to avoid misunderstanding about the use of probiotics, as we considered only patients with AR, excluding prevention of AR onset.
The paper then moves to 4 new studies on the treatment of AR. Only one is of a suitable size. One paper of 28 patients..
- Torre, E.; Sola, D.; Caramaschi, A.; Mignone, F.; Bona, E.; Fallarini, S. A Pilot Study on Clinical Scores, 320 Immune Cell Modulation, and Microbiota Composition in Allergic Patients with Rhinitis and 321 Asthma Treated with a Probiotic Preparation. Int. Arch. Allergy Immunol. 2022, 183, 186-200
Makes very strong immunological findings which had not been found in many other studies that the authors have highlighted. Specifically that probiotics reduced the peripheral eosinophils, the levels of type 2 cytokines IL-4 and IL- 5 in addition to the changes in clinical scores. This needs some commentary about the fact that this is a small study. No comment was made about the timing of this study. It would be important to know if immunological differences correlated with clinical symptoms.
R Many thanks for this comment. We expanded the discussion considering this issue.
In the final statements
“The actual literature evidence reasonably seems to suggest that probiotics could be a profitable add-on strategy in managing patients with AR. However, no definitive recommendations can still be defined now.”
I think a better word than “profitable” could be used.
R Many thanks for this comment. We changed the word as suggested.
The mechanisms picture is nice, but there is only data from small studies to support the immunological changes that is still not convincing.
R Many thanks for this comment. We outlined this concept in the text.
As a comment
There are many co-factors that might influence probiotics in any disease such as antibiotic use, PPIs, diet (fibre, lipids, fast food, protein, fructose) and even stress. Add on top of this, SNPs in Toll4 receptors and other bacterial ligands like TRPA1, there is always going to massive heterogeneity. In probiotic studies, it may be more beneficial to look at responders and non-responders to better elucidate how and if these agents are working. Further to this we have strain variation as well as number of bacteria and viability of the probiotics used
The authors need not include those thoughts and indeed, as an expert in the area - I am sure Professors Ciprandi and Tosca could have many other factors to add.
R Many thanks for this comment. We expanded the discussion considering these suggestions.
Overall – as mentioned it is an update from a highly respected author.
R Many thanks for this comment.
I would like to see commentary about possible papers that the recent reviewers missed.
R As specified in the text, we included all studies published in the last year.
The review answers thew question posed in the title.
R Many thanks!
Reviewer 3 Report
This is an exhaustive review of the latest updates on the problem. Probiotics could offer a new option in medical therapy of AR apart from antihistamines, corticosteroids, and ASIT.
Author Response
This is an exhaustive review of the latest updates on the problem. Probiotics could offer a new option in medical therapy of AR apart from antihistamines, corticosteroids, and ASIT.
R Many thanks for this comment.
Round 2
Reviewer 1 Report
N/A
Author Response
It seems that the Reviewer had no comment.